# Increased Tolerance of *Massion’s pine* to Multiple-Toxic-Metal Stress Mediated by Ectomycorrhizal Fungi

**DOI:** 10.3390/plants12183179

**Published:** 2023-09-05

**Authors:** Taoxiang Zhang, Panpan Zhang, Wenbo Pang, Yuhu Zhang, Hend. A. Alwathnani, Christopher Rensing, Wenhao Yang

**Affiliations:** 1International Joint Laboratory of Forest Symbiology, College of Forestry, Fujian Agriculture and Forestry University, Fuzhou 350002, China; ning527@fafu.edu.cn (T.Z.); zpanpan0819@163.com (P.Z.); 18731112578@163.com (W.P.); zhangyuhu4790@163.com (Y.Z.); 2Key Laboratory of Soil Ecosystem Health and Regulation of Fujian Provincial University, College of Resources and Environment, Fujian Agriculture and Forestry University, Fuzhou 350002, China; 3210422053@fafu.edu.cn (H.A.A.); 3210422089@fafu.edu.cn (C.R.)

**Keywords:** *Cenococcum geophilum* (*C. geophilum*), multiple-toxic-metal stress, *Massion’s pine*, transcriptomic analysis

## Abstract

*Pinus massoniana (Massion’s pine)*, a pioneer tree species, exhibits restoration potential in polluted mining areas. However, the physiological and molecular mechanisms of ectomycorrhizal (ECM) fungi in *Massion’s pine* adaptability to multiple-toxic-metal stress are still unclear. Hence, *Massion’s pine* seedlings inoculated with two strains of *C. geophilum*, which were screened and isolated from a polluted mine area, were cultivated in mine soil for 90 days to investigate the roles of EMF in mediating toxic metal tolerance in host plants. The results showed that compared with the non-inoculation control, *C. geophilum* (CG1 and CG2) significantly promoted the biomass, root morphology, element absorption, photosynthetic characteristics, antioxidant enzyme activities (CAT, POD, and SOD), and proline content of *Massion’s pine* seedlings in mine soil. *C. geophilum* increased the concentrations of Cr, Cd, Pb, and Mn in the roots of *Massion’s pine* seedlings, with CG1 significantly increasing the concentrations of Pb and Mn by 246% and 162% and CG2 significantly increasing the concentrations of Cr and Pb by 102% and 78%. In contrast, *C. geophilum* reduced the concentrations of Cr, Cd, Pb, and Mn in the shoots by 14%, 33%, 27%, and 14% on average, respectively. In addition, *C. geophilum* significantly reduced the transfer factor (TF) of Cr, Cd, Pb, and Mn by 32–58%, 17–26%, 68–75%, and 18–64%, respectively, and the bio-concentration factor (BF) of Cd by 39–71%. Comparative transcriptomic analysis demonstrated that the differently expressed genes (DEGs) were mainly encoding functions involved in “transmembrane transport”, “ion transport”, “oxidation reduction process”, “oxidative phosphorylation”, “carbon metabolism”, “glycolysis/gluconeogenesis”, “photosynthesis”, and “biosynthesis of amino acids.” These results indicate that *C. geophilum* is able to mitigate toxic metals stress by promoting nutrient uptake, photosynthesis, and plant growth, thereby modulating the antioxidant system to reduce oxidative stress and reducing the transport and enrichment of toxic metals from the root to the shoot of *Massion’s pine* seedlings.

## 1. Introduction

Mining activity is one of the major sources of soil heavy metal contamination [1], which has been regarded as a major environmental hazard resulting in forest decline [2]. The high concentrations of heavy metals in mining areas have been shown to cause changes in plant physiology and metabolic activities, such as inhibition of chlorophyll biosynthesis, absorption of nutrients and water, photosynthetic rate, carbon fixation, root development, etc., resulting in a scarcity of plants in the mining area and seriously affecting the stability of forest ecosystems [3,4,5,6]. Therefore, it is essential to overcome such environmental hazards and develop efficient strategies of plant protection for reforestation in mining areas [2]. However, it should be noted that the toxicity of metals, low nutrients, and bad soil structure in mining areas greatly restrict the growth of ecological restoration tree species and thereby make restoration difficult [7,8]. Thus, reforestation is dependent on the use of suitable tree species to cope with the local soil’s environmental constraining factors.

Ectomycorrhiza (ECM) is a mutualistic symbiosis formed between ectomycorrhizal (ECM) fungi and the roots of higher plants [9]. ECM fungi can provide mineral nutrients to the plant, and the plant, in turn, supplies the fungus with photosynthetically derived carbohydrates [10,11]. Under natural conditions, ectomycorrhizal fungi are able to colonize the roots of a wide range of woody plants, such as *Eucalyptus*, *Pinus*, *Acacia*, and *Picea* [11]. Ectomycorrhiza have been shown to improve the survival of plants under various stresses, including heavy metals, by promoting the absorption capacity of plants to water and nutrients, improving plant antioxidant activity and intracellular metal sequestration, increasing the root area and plant nutrient uptake, etc. [9,12,13,14]. In addition, ectomycorrhiza are able to play the role of a “barrier” to prevent the transport of heavy metals to the aerial parts of host plants through their own typical Hartig network structure and fungus sheath, and they can enhance the tolerance of plants to heavy metal stress [15]. Previous studies have shown that different heavy metals accumulate in different tissue parts of the fungi, with the Hartig net being the main site for cadmium accumulation, while zinc is mainly accumulated in the cell wall and cytoplasm of mycorrhizal roots, reducing the transport of heavy metals to spruce roots [16,17]. However, some studies have shown that ECM can significantly promote plant uptake of heavy metal, with higher heavy metal content in the aboveground parts of mycorrhizal plants compared to non-mycorrhizal plants [2,10,12]. Moreover, high concentrations of heavy metal stress do not significantly inhibit plant growth [8,18].

*Pinus massoniana (Massion’s pine)* is a fast-growing and high-yielding tree species that has been widely planted in deforested areas in southern China [19]. It displays various characteristics, such as strong adaptability, rapid growth, and low reforestation costs, and it is one of the ideal pioneer tree species for the restoration of polluted mining areas [7,20]. Studies have shown that the symbiotic relationship between *Massion’s pine* root and most ectomycorrhizal fungi species is one of the main reasons for its increased survival in polluted soil [2]. Previous field studies have reported that *Massion’s pine* trees colonized with ECM fungi were able to grow better in metal-polluted mine soils [7,21]. The application of ECM fungi in abandoned mining areas appears to be crucial for improving the survival and growth of seedlings under local stressful site conditions [22]. However, the physiological and molecular responses of *Massion’s pine* seedlings inoculated with *C. geophilumi* to multiple-toxic-metal stress are still unclear.

In this study, the contribution of EMF (two strains of *C. geophilum* isolated from a polluted mining area) on the growth, toxic metal accumulation, absorption of mineral nutrients, antioxidant capacity, and related gene expression of *Massion’s pine* exposed to a multiple-metal-contaminated soil was investigated. The objectives were to reveal the physiological and molecular mechanisms of multiple metal tolerance of *Massion’s pine* seedlings mediated by *C. geophilum*. This study expanded the understanding of the role of ECM fungi in plants to multiple-toxic-metal stress and provided a feasible strategy for assisting the reforestation of mining sites by ectomycorrhizal plants.

## 2. Results

### 2.1. The Effects of C. geophilum on Biomass, Root Morphology, and Element Absorption of Massion’s pine Seedlings in Mine Soil

The results of *Massion’s pine* seedling height, biomass, and root morphology are presented in Figure 1 and Table 1. Compared to the non-inoculation (NM) seedling, *C. geophilum* significantly increased the plant height, shoot fresh weight, root fresh weight, shoot dry weight, and root dry weight of *Massion’s pine* seedlings by 15–25%, 40–58%, 125–136%, 32–52%, and 28–39%, respectively. The results showed that CG1 and CG2 could promote the plant biomass under toxic metal stress. As shown in Table 1, the root surface area, average root diameter, and root volume of *Massion’s pine* seedlings inoculated with *C. geophilum* were significantly higher than in NM seedlings (*p < 0.05*), indicating that *C. geophilum* can effectively promote the root development and improve the tolerance of *Massion’s pine* to toxic metals. We determined the concentrations of nutrient elements (N, P, and K) in the roots and shoots of *Massion’s pine* seedlings (Figure 2A–C). The results showed that, compared to NM seedlings, ECM fungi significantly increased the concentrations of N, P, and K in the shoots and roots of *Massion’s pine* seedlings.

To investigate the effect of *C. geophilum* on toxic metal accumulation, we measured Cr, Cd, Pb, and Mn content in roots and shoots (Figure 3). Compared to NM seedlings, Cr, Cd, Pb, and Mn uptake was significantly different in inoculated plants. CG1 and CG2 treatment increased the concentrations of Cr, Cd, Pb, and Mn in the roots of *Massion’s pine* seedlings. CG1 significantly increased the concentrations of Pb and Mn in the roots by 246% and 162%, while CG2 significantly increased the concentrations of Cr and Pb by about 102% and 78% (Figure 3B). In contrast, both CG1 and CG2 reduced the concentrations of Cr, Cd, Pb, and Mn in the shoots by 14% (11–17%), 33% (26–39%), 27% (11–43%), and 14% (6–21%) on average, respectively, and the reduction of Mn, Cd, and Pb in the shoots under CG2 treatment was significant (Figure 3A, *p* < 0.05).

The transfer factor (TF) and bioconcentration factor (BF) of plants were analyzed to clarify the impact of *C. geophilum* on the transport of toxic metals from soil/root to shoot of *Massion’s pine* (Figure 3C,D). The results showed that CG1 and CG2 treatments significantly reduced the TF of *Massion’s pine* (*p < 0.05*). Compared to NM, the transport of Cr, Cd, Pb, and Mn from root to shoot in CG1- and CG2-inoculated seedlings was reduced by 32%, 26%, 75%, and 64% and by 58%, 17%, 68%, and 18%, respectively (Figure 3C). In addition, CG1 significantly reduced the BF of Cd by 39%, while CG2 significantly reduced the BF of Cd, Pb, and Mn by 71%, 40%, and 22%, respectively (Figure 3D), indicating that *C. geophilum* was able to effectively reduce the transport and enrichment of toxic metals from the soil and root to the shoot, thus alleviating the stress of toxic metals on *Massion’s pine* seedlings.

### 2.2. Effects of C. geophilum on Photosynthetic Characteristics and Antioxidant Enzyme Activities of Massion’s pine Seedlings in Mining Soil

As shown in Figure 4A–D, *C. geophilum* promoted the photosynthesis of *Massion’s pine* seedlings in mining soil. Compared to NM seedlings, CG1 and CG2 treatments significantly increased the net photosynthetic rate (Pn), transpiration rate (Tr), and stomatal conductance (Gs) of *Massion’s pine* seedlings by 6.4–8.7 times, 1.7–3.7 times, and 3.9–4.9 times, and reduced intercellular CO_2_ (Ci) by 36% and 37%.

The accumulation of malondialdehyde (MDA) in the shoot and root of CG1 and CG2 treatment was significantly reduced, with the shoot and root MDA content being reduced by 20–23% and 28–29%, indicating that inoculation with *C. geophilum* reduced the damage of toxic metals to plant cell membranes (Figure 5A). No significant differences in MDA content in shoots were observed between CG1 and CG2 treatments.

*C. geophilum* substantially increased proline content in shoots and roots of *Massion’s pine* seedlings under toxic metal stress. The content of proline in the shoot was higher than that in the root. The proline content increased significantly by 159% and 174% in roots under CG1 and CG2 treatment, while the proline content in the shoot increased by 16% and 20% (Figure 5B).

The results showed that compared to the NM seedlings, CG1 and CG2 treatment promoted the activities of antioxidant enzymes CAT, POD, and SOD in the shoots of *Massion’s pine* seedlings by 60% (51–69%), 225% (177–273%), and 9% (7–11%) and in the roots by 168% (166–170%), 200% (165–235%), and 11% (10–11%) on average, respectively. Hence, *C. geophilum* was able to enhance the ability of seedlings to resist toxic metal stress by stimulating CAT, POD, and SOD activities (Figure 6A–C).

### 2.3. Validation of Differentially Expressed Genes (DEGs)

A total of 55.83 GB of clean data was obtained from the RNA-Seq analysis, with all samples achieving 4.25 GB of clean data and 94.06% of Q30 base percentages (Appendix A). After assembly, a total of 38,165 unigenes were obtained, with an N50 of 2354, indicating high assembly integrity (Appendix A). Fold Change ≥ 2 and FDR < 0.05 were used as screening criteria for differentially expressed genes (DEG). Volcano maps (Figure 7A,B, respectively) represent the expression of DEGs in CG1 and CG2 treatments in mine soil. Specifically, compared with NM, there were 3705 up-regulated and 4558 down-regulated DEGs under CG1 treatment. In addition, CG2 showed 4540 up-regulated and 3907 down-regulated DEGs (Figure 7). Between the CG1 and CG2 treatments, there were shared 2777 up-regulated DEGs and 2768 down-regulated DEGs (Figure 8A,B).

To understand the function of DEGs in *Massion’s pine*, a gene ontology term enrichment analysis (GO) was performed. Go enrichment analysis showed that DEGs of CG1 and CG2 treatments were enriched in some pathways of molecular function, cellular component, and biological processes (Figure 9A,B). The top GO items “transmembrane transport (GO:0055085)”, “ion transport (GO:0006811)”, “transport (GO:0006810)”, “monocarboxylic acid metabolic process (GO:0032787)”, and “oxidation reduction process (GO:0055114)”, which belonged to the biological processes, “organelle membrane (GO:0031090)”, “membrane (GO:0016020)”, and “vacuolar (GO:0005773)”, which belonged to the cellular component category, as well as “transmembrane transporter activity (GO:0022857)” and “transporter activity (GO:0005215)”, which belonged to the molecular function category, were all enriched under CG1 and CG2 treatments.

To study the biological behavior of the DEGs involved in the toxic metal stress response, we mapped the DEGs to the reference canonical pathways in the Kyoto Encyclopedia of Genes and Genomes (KEGG) to further identify the regulatory pathways mediated by *C. geophilum* (Figure 10A,B). CG1 and CG2 shared the pathways of “oxidative phosphorylation”, “carbon metabolism”, “glycolysis/gluconeogenesis”, “photosynthesis”, and “biosynthesis of amino acids”, indicating that nutrient and energy utilization, which is an important response mechanism of *Massion’s pine* seedlings to toxic metals stress, mediated by *C. geophilum* occurred.

We created a heatmap of part up-regulated DEGs enriched in the same pathway to select the key genes of *Massion’s pine* conferring resistance to heavy metal stress mediated by *C. geophilum* (Figure 11). There are 78 up-regulated DEGs enriched in the common pathway of oxidative phosphorylation, 72 up-regulated DEGs enriched in carbon metabolism, 39 DEGs enriched in glycolysis/gluconeogenesis, 7 DEGs enriched in photosynthesis (Figure 11A–D), 36 up-regulated DEGs enriched in ion transport (GO:0006811), and 34 up-regulated DEGs enriched in transporter activity (GO:0005215) (Figure 11E,F).

### 2.4. RT-qPCR Verification of DEGs

In order to verify the reliability of the sequencing results, we selected one gene in the carbon metabolism pathway (Figure 12A), two genes encoding functions in the oxidative phosphorylation pathway (Figure 12B,C), three genes in the ion transport (Figure 12D–F), and two genes in the transporter activity (Figure 12G,H) for real-time quantification polymerase chain reaction (RT-qPCR) validation. The relative gene expression of RT-qPCR was calculated using the 2^−∆∆Ct^ method. RT-qPCR analysis showed that CG1 and CG2 significantly upregulated the eight genes compared to NM, which is consistent with the results of RNA-Seq analysis (Figure 12). The results indicate that reliable RNA-Seq data were obtained.

## 3. Discussion

Generally, the unfavorable edaphic conditions in the soil of mine areas (e.g., toxic metal pollution, low pH, poor physical structure, low nutrition availability) greatly limit plant growth required for restoration [13,23]. Heavy metal contamination of soil has been shown to inhibit plant growth by reducing nutrient uptake by plants [17]. It has also been frequently reported that ECM symbiosis is able to enhance the transfer of nutrients from the soil to their host plants under HM stress [6,24]. For example, Liu et al. (2020) [15] showed that inoculation with *Pisolithus* sp1. and *Laccaria* sp. increased the nutrient concentrations in the host plants *Pinus sylvestris*. Similarly, higher levels of N, P, Fe, and Mg were found in *Pinus sylvestris* seedlings colonized by the *Suillus bovinus* [25]. Thus, forming stable mycorrhizal symbiosis with the roots of host plants, improving plant nutrition, and enhancing plant growth in metal-contaminated soils have been demonstrated to be major functions of the ECM symbiosis [13,26]. In this study, we found that ECM fungi inoculation promoted plant growth by increasing seedling heights, biomass, and root parameters (Figure 1 and Table 1). This was probably due to the increased root area for better water and mineral nutrient uptake [27]. Similarly, Dagher et al. (2020) [28] found that willows (*Salix* sp.) inoculated with the ECM fungus *Sphaerosporella brunnea* were able to generate significantly higher plant biomass. Root colonization by ectomycorrhizal fungi *Bovista limosa PY5* significantly promoted the growth and increased the biomass of *Populus yunnanensis* trees in a metal mine tailings pond [29]. The promoting effect of mycorrhizal fungi on plant growth and root development may be related to the increased nutrient absorption and better physiological status of plants [30]. Overall, in this research, the improved absorption of major nutrients and up-regulated DEGs enriched in “carbon metabolism” and “oxidative phosphorylation” may be an effective way for plants to cope with metal toxicity in mining soils.

With regard to the effect of ECM fungi on metal uptake and transport, our results showed that *C. geophilum* significantly increased the Pb concentrations in *Massion’s pine* roots by 78–246% (Figure 3B) but inhibited the translocation of Cr, Cd, Pb, and Mn from the roots to the shoots by 32–58%, 17–26%, 68–75%, and 18–64%, respectively (Figure 3C). These are consistent with some recent reports [31,32]. For example, inoculation with *Suillus luteus* increased the heavy metal concentrations in the roots of *Massion’s pine* but decreased it in the shoots [21]. Similar results were observed that ECM increased Cd accumulation in the roots of *Quercus acutissima* seedlings while reducing Cd accumulation in the leaves and stems [33]. The reduction in heavy metal concentration of the mycorrhizal host was more likely correlated to the promotion of shoot biomass, which might cause the dilution effect [34]. In addition, the heavy metals taken up by mycorrhizal fungi were restricted by the Casparian strips, which may also have inhibited the transport of HMs to the aboveground parts [31]. ECM can promote the uptake and transfer of essential nutrients to suppress the transport of heavy metals, which may be another reason for the reduction of HM concentration in the mycorrhizal host [24,26]. For example, the metal-tolerant ecotypes of *Suillus* species were reported to reduce the transfer of metals in the direction of the plant–fungus interface without hampering nutrient transfer to the host plant [26]. Kong et al. (2020) [24] reported that enrichment of N and P significantly decreased the upward Cd transfer and enhanced the root uptake. In our study, the CG1 and CG2 both promoted the transfer of N, P, and K to the leaves (Figure 2), which might have contributed to the decreased accumulation of heavy metals. In addition, the low level of metal transporter may also help to reduce the heavy metal concentrations in ECM-fungi-colonized plants [24]. Some previous studies have demonstrated that ECM was able to fix Cd in the roots by cell wall binding and cytoplasmic chelation to reduce metal damage to aboveground parts [32,35]. We found that GO enrichment analysis of DEGs showed enrichment mainly in “transmembrane transport”, “ion transport”, “transport”, “organelle membrane”, “membrane”, “vacuolar”, “transmembrane transporter activity”, and “transporter activity” items (Figure 9). Therefore, combined with the concentrations of Cr, Cd, Pb, and Mn in roots and shoots (Figure 3A,B), the decrease of the TF values (Figure 3C), and the increase of DEGs expression in membrane transport (Figure 9 and Figure 11E,F), we speculate that *C. geophilum* can bind and transport Cr, Cd, Pb, and Mn through ion and membrane transport on the cell membrane, decreasing the upward transfer of toxic metals to plants and reducing the content of toxic metals in shoots.

Plants growing under heavy metal stress usually display inhibition of photosynthesis due to chlorophyll degradation or reduced biosynthesis [36,37]. It has been well demonstrated that mycorrhizal symbiosis has positive effects on photosynthesis [13]. Our results also showed that CG1 and CG2 significantly increased the photosynthetic efficiency of *Massion’s pine* (Figure 4), and DEGs were enriched in the phosphorylation pathway and significantly up-regulated in CG1 and CG2 seedlings (Figure 10). Previous research has demonstrated that the enhanced host plant photosynthesis due to ectomycorrhizal fungi was correlated to the increase in belowground carbon demand [36]. Nitrogen has been considered as a key factor regulating photosynthesis because it is a major component of photosynthetic enzymes and structures [38,39]. In addition, in our study, the CG inoculation resulted in a significant increase in leaf N, P, and K concentrations (Figure 2) and significantly enriched “oxidative phosphorylation” and “carbon metabolism” pathways, which may be linked to higher photosynthetic activity of *Massion’s pine* inoculated by *C. geophilum* (Figure 10).

It has been well documented that heavy metal affects various plant physiological processes [40,41]. The production of reactive oxygen species (ROS) has been shown to be a common feature of plants under metal stress [14,42]. Plants are able to increase their antioxidant activities for defending against abiotic and biotic stress, and mycorrhizal symbiosis has been shown to further enhance the activities of antioxidant enzymes to mitigate H_2_O_2_ and lipid peroxidation damage [13,35]. For example, biotic stress caused by *Botrytis cinerea* in lettuce induced the activity of CAT, SOD, and glutathione [43], while *Fusarium culmorum* in wheat induced the activity of CAT and POD [44]. *Suillus luteus* increased the activities of CAT, APX, and SOD in leaves of *Quercus acutissima* under Cd stress [45], while *Bovistalmosa* triggered the antioxidant defense of *Populus yunnanensis* by enhancing the activities of SOD and CAT [29]. Consistent with this work, our results also revealed that CG1 and CG2 significantly increased the antioxidant enzyme CAT, POD, and SOD activities in the shoot of *Massion’s pine* by 60%, 225%, and 9% and in the root by 168%, 200%, and 11% on average, respectively (Figure 6). This may be correlated to the up-regulated genes encoding functions involved with the antioxidant defense after fungal colonization [13]. Moreover, we also discovered that a large number of DEGs were significantly enriched in functions of oxidation reduction processes in the KEGG enrichment pathway (Figure 10) as a response to oxidative stress and cellular oxidant detoxification. These results indicate that ectomycorrhizal fungi may stimulate the expression of genes involved in defense against reactive oxygen species in *Massion’s pine* seedlings in order to enhance the activity of antioxidant enzymes and maintain ROS balance in *Massion’s pine*.

The level of MDA (a cytotoxic product of lipid peroxidation) is used to measure the state of lipid peroxidation and the degree of cell membrane damage induced by ROS production [46]. Studies have indicated that oxidative damage caused by heavy metal stress increased the MDA contents of plants [47,48]. In this study, the levels of MDA in the shoots and roots of ectomycorrhizal plants were lower than those of NM plants, suggesting that CG1 and CG2 seedlings had a greater ability to counteract oxidative stress by reducing membrane lipid peroxidation (Figure 5A). Proline has been shown to have an important role in enhancing antioxidant defense to alleviate metal toxicity [49]. In our study, the proline content increased significantly by 159% and 174% in the roots and increased by 16% and 20% in the shoots under CG1 and CG2 treatment (Figure 5B). ECM fungi were able to increase the absorption of N and P to synthesize proline [30]. Similar results were found in our study, where ectomycorrhizal fungi promoted the absorption of N, P, and K of host plants. Importantly, CG1 and CG2 treatments significantly enriched the “Proline synthesis” pathway in the KEGG enrichment pathway, and proline-synthesis-related genes were significantly up-regulated. Hence, we speculate that the enhanced proline content may be another mechanism of *C. geophilum* for alleviating heavy metal toxicity in the host plants. However, due to time constraints, our experiment mainly analyzes the impact of *C. geophilum* on the resistance of *Massion’s pine* seedlings to heavy metal stress. In future experiments, we will continue to conduct long-term toxic metal stress experiments in mature plants.

## 4. Materials and Methods

### 4.1. Preparation of Massion’s pine Seedlings Cultivation

Healthy and plump seeds of *Massion’s pine* provided by Wuyi Forest Farm, Zhangping, Fujian Province, were selected and sterilized using hydrogen peroxide (30%) for 10 min, followed by washing with ultra-pure water 5–6 times and then immersion in sterilized water for 12–24 h. The seeds were then germinated in a seedling tray containing autoclaved vermiculite (121 °C, 3 h) at 25 °C in an incubator and cultured in a greenhouse (25 °C, relative humidity ranging from 65% to 70%, at 300 μmol (m^−2^ s^−1^) light intensity with a 16 h light/8 h dark regime, respectively) for 30 days. After sprouting, the seedlings were transplanted in a rectangular plastic petri dish (23.5 × 8.5 × 1.6 cm, 3 plants per box) containing mixed substrates of forest soil and Akadama clay at a ratio of 1:1 (autoclaved at 121 °C for 3 h) and then placed in a greenhouse under the same conditions for 2 weeks.

### 4.2. Inoculation of Massion’s pine Seedlings

Two strains of *Cenococcum geophilum* (*C. geophilum*) (CG1 and CG2), an ectomycorrhizal fungal species isolated from *Massion’s pine* rhizosphere in a polluted mining area by Fujian Agriculture and Forest University, were used in this study. The two strains (CG1 and CG2) were pre-cultured in modified Melin–Norkrans (MMN) agar medium at 25 °C in the dark for 45 days. The healthy *Massion’s pine* seedlings with consistent growth were inoculated with the same amount of fungal blocks, and the control was inoculated with the same amount of MMN agar medium. The inoculated seedlings were cultured in the greenhouse (25 °C, light 18 h, dark 6 h) for 2 months to form mycorrhizal seedlings for experiments and alternately irrigated with 200 mL of distilled water and 1/1000 Hoagland nutrient solution once a week. Nine replicates of each treatment were conducted. After growing for 3 months in a greenhouse, the roots of *Massion’s pine* inoculated with and without *C. geophilum* were observed under the stereo microscope to check the mycelium colonization on the roots (Appendix A).

### 4.3. Soil Collection and Characterization

Three replicates were sampled from the surface horizon (0–20 cm), and each sample was a composite of five individual soil cores taken at 5 m intervals. After removing stones and discrete plant residues, soil samples were air dried and sieved through a 2 mm stainless steel mesh. One part of the soil was sterilized at 120 °C for 70 min three times to eliminate competing fungal species. Another part of the soil was used for physical and chemical analyses. Soil pH, soil organic matter (SOM), total potassium (TK), and total phosphorus (TP) were determined by the methods described by the Agricultural Chemistry Committee of China. In addition, 0.1 g of soil samples was digested with 10 mL acidic mixture (HF/HClO_4_/HNO_3_; 7:2:1 *v*/*v*/*v*) on a hotplate at 220 °C [50]; then, the concentration of cadmium (Cd), manganese (Mn), chromium (Cr), and lead (Pb) in the digested solution was determined by an inductively coupled plasma emission spectrometer (ICP-OES, Avio 200, Perkin Elmer, Waltham, MA, USA) (Table 2).

The greenhouse pot experiment included the ECM fungal treatments (non-inoculation control (NM) and CG1 and CG2). Each treatment was replicated nine times. *Massion’s pine* with 90% mycorrhizal infection rate was transferred to plastic pots (13.5 × 9.5 cm) containing 400 g of sterilized mining soil. Three mycorrhizal plants were planted in each pot. Subsequently, the seedlings of each treatment were cultured in a greenhouse (25 °C, light 18 h, dark 6 h) and regularly irrigated with 70 mL sterilized water once a week. The seedlings were collected after 90 days of multiple-toxic-metal stress in the mining soil.

After three months of multiple-toxic-metal stress, the harvested seedlings were carefully cleaned and divided into shoots and roots. One part was immediately frozen in liquid nitrogen and stored at −80 °C until it was used for physiological measurements and RNA-Seq. The other part was used to analyze fresh biomass, plant height, and root morphology, and then it was dried at 60 °C for 72 h to obtain the constant weight to determine the dry weight. Rhizosphere soil was air dried and passed through 0.149 mm and 2 mm sieves for the determination of the total toxic metal content and the soil’s physical and chemical properties, respectively.

### 4.4. Measurement of Morphological Parameters

Three plants were randomly selected from nine replicates of each treatment. The roots of the seedlings were gently shaken to remove the soil from the root surface and carefully washed with running water. The morphological indices, such as total root length, root surface area, root average diameter, root volume, and root tip number, were analyzed and measured by Espon Expression 11000XL scanner and Win RHIZO software (Regent, QC, Canada). Shoot and root fresh weight and dry weight were recorded by an electronic balance.

### 4.5. Determination of Photosynthetic Characteristics, Antioxidant Systems, and Osmotic Regulating Substances in Massion’s pine Seedlings

A Li-6400XT portable photosynthetic system (Li-COR, Lincoln, NE, USA) was used to measure the photosynthetic gas exchange characteristics of the *Massion’s pine* seedlings. Then, 0.2 g of shoot and root tissue with three replications was ground in a mortar with liquid nitrogen for the determination of physiological indices. Next, 0.2 g of leaf and root samples was extracted with 0.05 M of sodium phosphate buffer (PBS, PH7.0) and centrifuged at 15,000× *g* for 15 min at 4 °C. The supernatant (enzyme extract) was used to determine the activity of peroxidase (POD), catalase (CAT), superoxide dismutase (SOD), and malondialdehyde (MDA). The SOD activity was analyzed by Microplate Reader (SpectraMax iD5, Molecular Devices, SV, USA) at 560 nm [51], the POD activity was measured at 470 nm [45], while CAT was measured at 240 nm [52]. The MDA concentration was determined by the thiobarbituric acid method, while the absorbance of the supernatant was read at 532 nm and 600 nm against a reagent blank [51].

The proline content was determined by [53]. Briefly, 0.2 g of leaf and root samples was extracted with 3% sulfosalicylic acid. An equal amount of glacial acetic acid and ninhydrin reagent was added to 2 mL of the extract and stored in boiling water for 1 h. After cooling, toluene was added, and chromophores were extracted from the aqueous phase. The absorbance was measured by Microplate Reader (SpectraMax iD5, Molecular Devices, Silicon Valley, NC, USA) at 520 nm and calculated as µmol g^−1^ DW relative to standard proline.

### 4.6. Determination of Elemental Content

Next, 0.2 g (dry weight) of ground shoot and root tissues was digested with 5 mL of mixed solution of nitric acid (AR) and hydrogen peroxide (30%) (*v*/*v*, 4:1) at 180 °C. The digestion liquid was used to measure the content of manganese (Mn), chromium (Cr), cadmium (Cd), lead (Pb), potassium (K), and phosphorus (P) using an inductively coupled plasma emission spectrometer (ICP-OES, Avio 200, Perkin Elmer, USA). In addition, 5 mL of concentrated sulfuric acid was used to digest the shoots and roots of plants at 220 °C to determine the total nitrogen (N) content by flow analyzer (San++, Skalar, Holland).

Bio-concentration factor (BF) is the ratio of the toxic metal concentration of the plant to the same element in the soil, and it is commonly used to measure the potential of plants for extracting heavy metals [54]. Transfer factor (TF) is an indicator of toxic metal translocation from root to shoot. These indicators were calculated as follows [55]:$$BF=\frac{C_{shoot}}{C_{soil}}$$
$$TF=\frac{C_{shoot}}{C_{root}}$$
where C_shoot_ and C_root_ are the toxic metal concentrations (mg/kg, dry weight) in the shoot and root, respectively. C_soil_ is the toxic metal concentration (mg/kg, dry soil) in the soil.

### 4.7. RNA-Seq Analysis

The RNA of *Massion’s pine* seedlings roots was extracted by CLB+Adlai RN40 reagent (RN40, Adlai) following the manufacturer’s instructions. RNA quantity and purity were estimated through the Nano Drop 2000 spectrophotometer (Thermo Fisher Scientific, Waltham, MA, USA), and the integrity was verified by Agilent 2100 Bio-analyzer (Agilent, Santa Clara, CA, USA). Construction of the cDNA library, quality verification, and further sequencing were performed by Biomarker Technologies (Beijing, China) using an Illumina Nova-Seq 6000 platform in accordance with standard protocols.

The raw data were filtered to obtain high-quality clean data, and the clean data were assembled using Trinity software to obtain the Unigene library of the *Massion’s pine* root. DEGs were identified with the DESeq R package (version 1.10.1). Gene expression was normalized to RPKM (reads per kb per million reads). The significant DEGs were defined based on a fold change ≥2 and a false discovery rate (FDR) < 0.05 and then subjected to an enrichment analysis using the GO functions and KEGG pathways.

### 4.8. Validation by RT-qPCR

In order to confirm the gene expression levels obtained from the RNA-Seq, an RT-qPCR analysis was performed. The gene AQP (Aquaporin protein gene) was used as the internal control, and specific primer sequences for selected genes were designed with Premier 5.0 (http://www.premierbiosoft.com/; accessed on 2 June 2022), which are presented in Appendix A.

### 4.9. Statistical Analysis

SPSS (Statistical Product and Service Solutions 26.0) software was used for statistical analysis. Under the condition of satisfying normality and homogeneity of variance, one-way analysis of variance (ANOVA) and Duncan’s test were used to evaluate the data at a significance level of *p* < 0.05. Data visualization was performed using Origin 2022b.

## 5. Conclusions

In the current study, we found that the ECM fungus *C. geophilum* effectively mitigated toxic-metal-induced toxicity in *Massion’s pine* seedlings by improving growth attributes, enhancing mineral element uptake, reducing the accumulation of Cr, Cd, Pb, and Mn in shoots, and increasing proline content and antioxidant enzyme activity. Our results suggested that applying the ECM fungus *C. geophilum* was an effective approach for alleviating multiple-metal-induced toxicity and improving the survival of *Massion’s pine* seedlings in mine soil. This study provided the theoretical foundation for applying ectomycorrhizal seedlings into the reforestation of mining sites.

## Figures and Tables

**Figure 1 plants-12-03179-f001:**
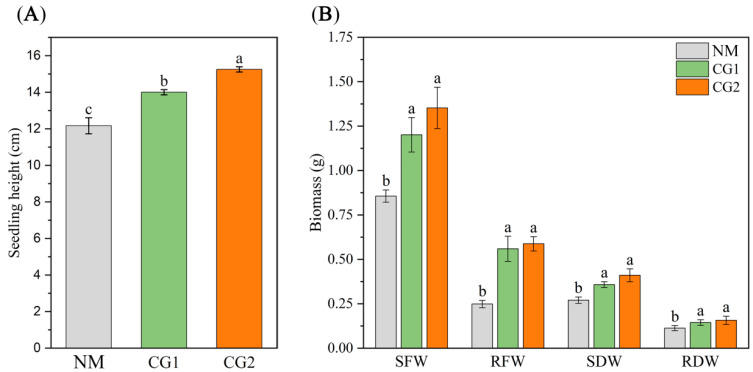
Effect of CG1 and CG2 inoculation on seedling height (**A**) and biomass (**B**) of *Massion’s pine* plants grown in toxic-metal-contaminated mine soil. Bars represent average values ± SE of three replicates. Different letters indicate statistically significant differences between different *C. geophilum* inoculations using one-way ANOVA followed by Dunnett’s test (*p* < 0.05). Non-inoculated plants (NM); shoot fresh weight (SFW); root fresh weight (RFW); shoot dry weight (SDW); root dry weight (RDW).

**Figure 2 plants-12-03179-f002:**
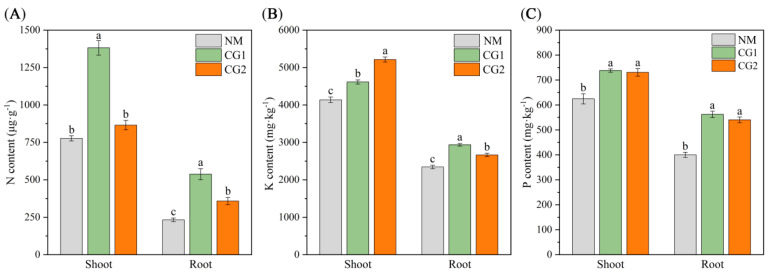
Nitrogen (N) (**A**), Potassium (K) (**B**), and Phosphorus (P) (**C**) content in shoots and roots of non-inoculated (NM) and inoculated (CG1, CG2) *Massion’s pine* plants grown in toxic-metal-contaminated mine soil. Bars represent average values ± SE of 3 replicates. Different letters indicate statistically significant differences between different *C. geophilum* inoculations by one-way ANOVA analysis followed by Dunnett’s test (*p* < 0.05).

**Figure 3 plants-12-03179-f003:**
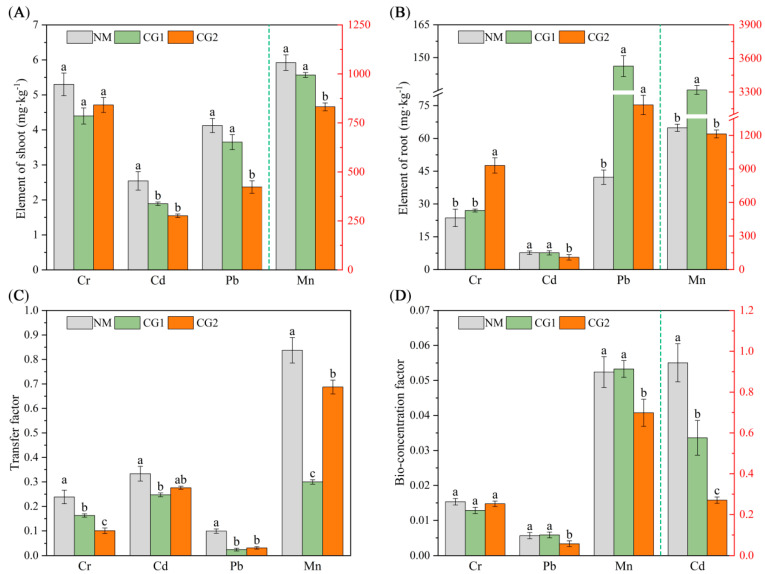
Cr, Cd, Pb, and Mn concentration in shoots (**A**) and roots (**B**), transfer factor (**C**), and bio-concentration factor (**D**) of non-inoculated (NM) and inoculated (CG1, CG2) *Massion’s pine* plants grown in toxic-metal-contaminated mine soil. Bars represent average values ± SE of 3 replicates. Different letters indicate statistically significant differences between different *C. geophilum* inoculations by one-way ANOVA analysis followed by Dunnett’s test (*p* < 0.05).

**Figure 4 plants-12-03179-f004:**
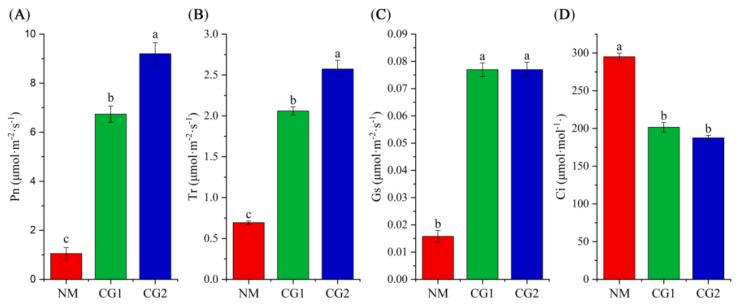
Net photosynthetic rate (Pn) (**A**), transpiration rate (Tr) (**B**), stomatal conductance (Gs) (**C**), and intercellular CO_2_ (Ci) (**D**) of non-inoculated (NM) and inoculated (CG1, CG2) *Massion’s pine* plants grown in toxic-metal-contaminated mine soil. Bars represent average values ± SE of 3 replicates. Different letters indicate statistically significant differences between different *C. geophilum* inoculations by one-way ANOVA analysis followed by Dunnett’s test (*p* < 0.05).

**Figure 5 plants-12-03179-f005:**
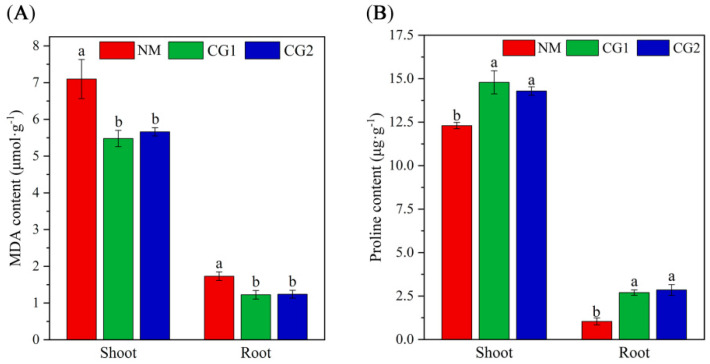
Malondialdehyde (MDA) (**A**) and proline (Pro) (**B**) content in shoots and roots of non-inoculated (NM) and inoculated (CG1, CG2) *Massion’s pine* plants grown in toxic-metal-contaminated mine soil. Bars represent average values ± SE of 3 replicates. Different letters indicate statistically significant differences between different *C. geophilum* inoculations by one-way ANOVA analysis followed by Dunnett’s test (*p* < 0.05).

**Figure 6 plants-12-03179-f006:**
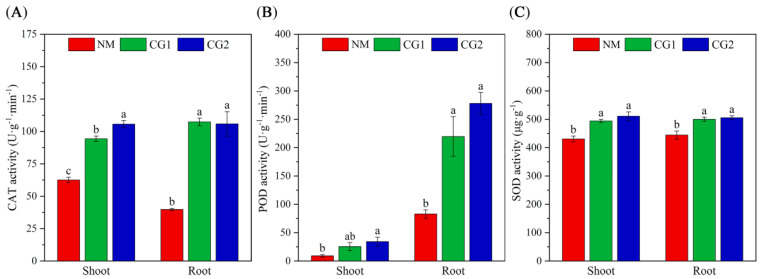
Catalase (CAT) (**A**), peroxidase (POD) (**B**), and superoxide dismutase (SOD) (**C**) activity in shoots and roost of non-inoculated (NM) and inoculated (CG1, CG2) *Massion’s pine* plants grown in toxic-metal-contaminated mine soil. Bars represent average values ± SE of 3 replicates. Different letters indicate statistically significant differences between different *C. geophilum* inoculations by one-way ANOVA analysis followed by Dunnett’s test (*p* < 0.05).

**Figure 7 plants-12-03179-f007:**
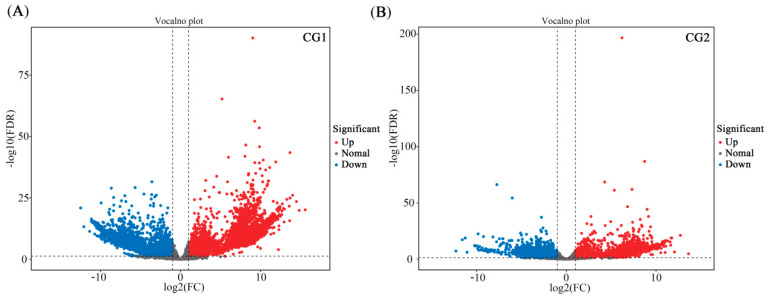
Volcano map of the candidate genes. Volcano map shows the differentially expressed genes (DEGs) of CG1 (**A**) and CG2 (**B**) inoculation compared to non-inoculation (NM) plants. Red spot indicates significantly up-regulated genes after CG1 and CG2 inoculation; blue spot indicates significantly down-regulated genes; and gray spot indicates no-changes genes. |log2Fold change| ≥ 2 and FDR < 0.05 were used as the screening criteria of DEGs.

**Figure 8 plants-12-03179-f008:**
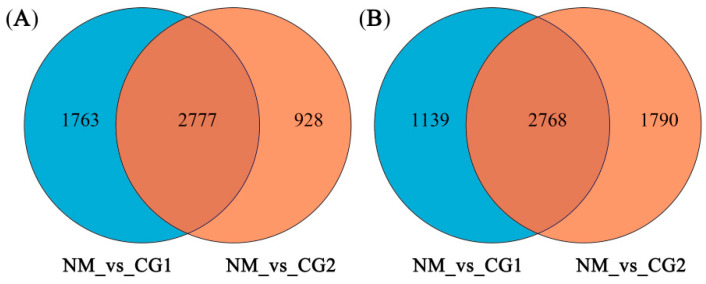
Venn diagram depicting the number of shared and specific differentially expressed genes (DEGs) induced by CG1 inoculation (NM_vs_CG1) and with CG2 inoculation (NM_vs_CG1). Up-regulated DEGs (**A**) and down-regulated DEGs (**B**). |log2Fold change| ≥ 2 and FDR < 0.05 were used as the screening criteria of DEGs. NM_vs_CG1/2 indicates: NM is the control group and CG1/2 is the experimental group.

**Figure 9 plants-12-03179-f009:**
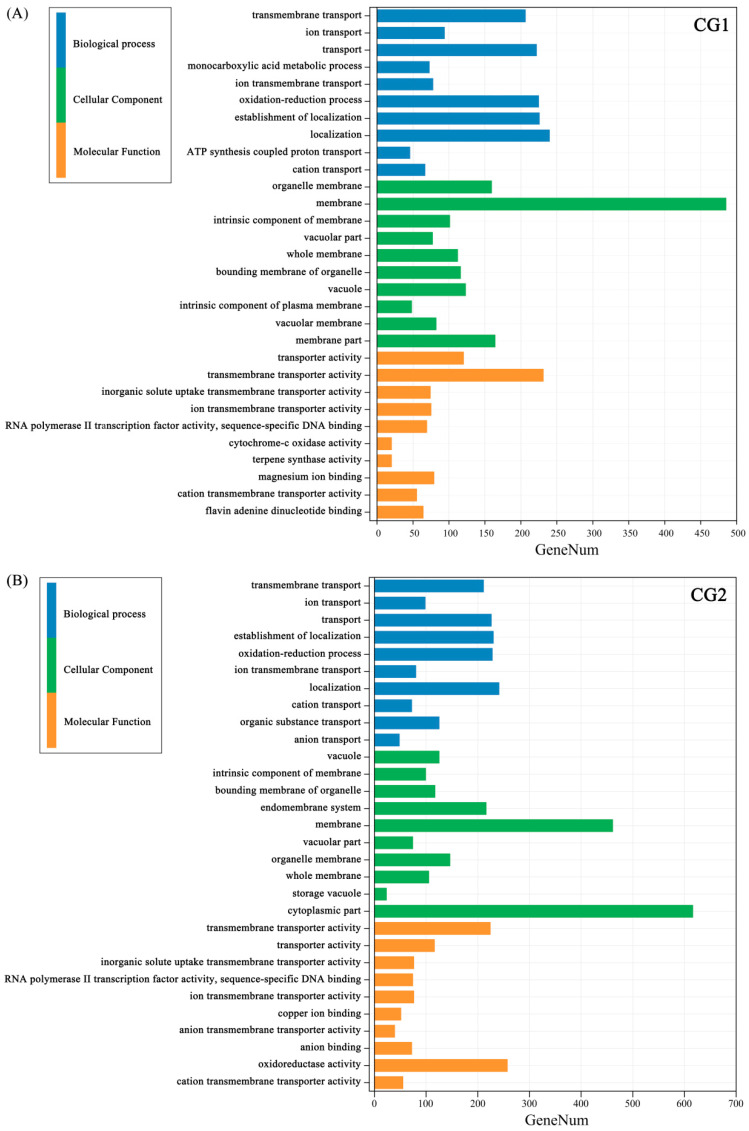
Gene ontology (GO) enrichment of the DEGs in CG1 (**A**) and CG2 (**B**) inoculation. The ordinate represents different regulatory pathways in the biological progress (blue), cellular component (green), and molecular function (yellow) categories, respectively. The abscissa represents the number of DEGs involved in the corresponding pathways.

**Figure 10 plants-12-03179-f010:**
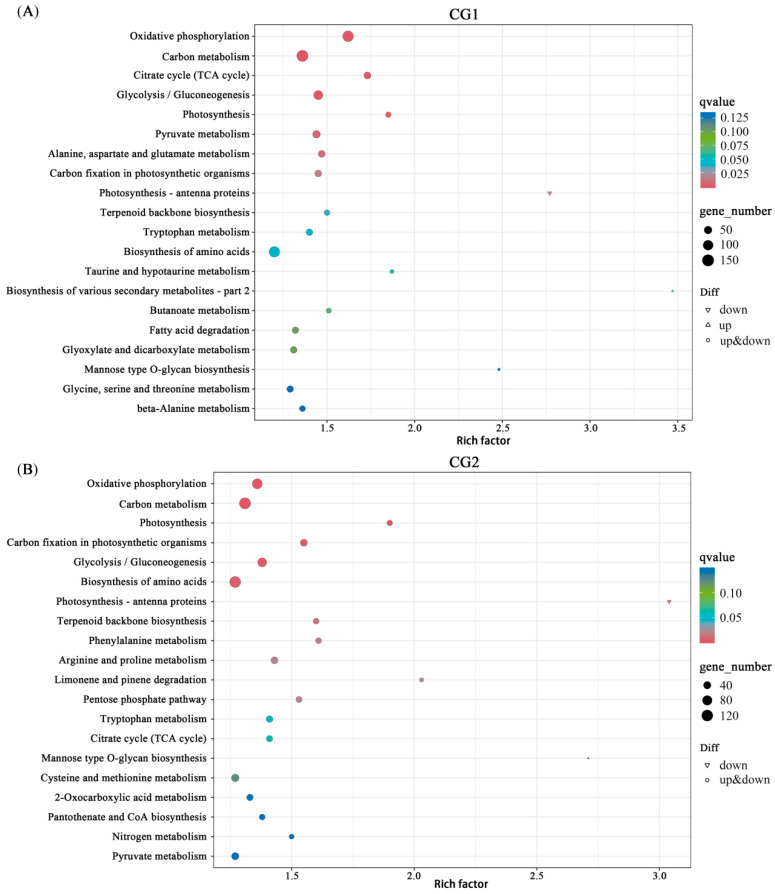
KEGG enrichment of the DEGs in CG1 (**A**) and CG2 (**B**) treatments. Each symbol (circle and triangle) in the figure represents a KEGG pathway, the ordinate indicates the name of the pathway, and the abscissa is the enrichment factor. The size of the symbol represents the number of DEGs involved in the corresponding pathways.

**Figure 11 plants-12-03179-f011:**
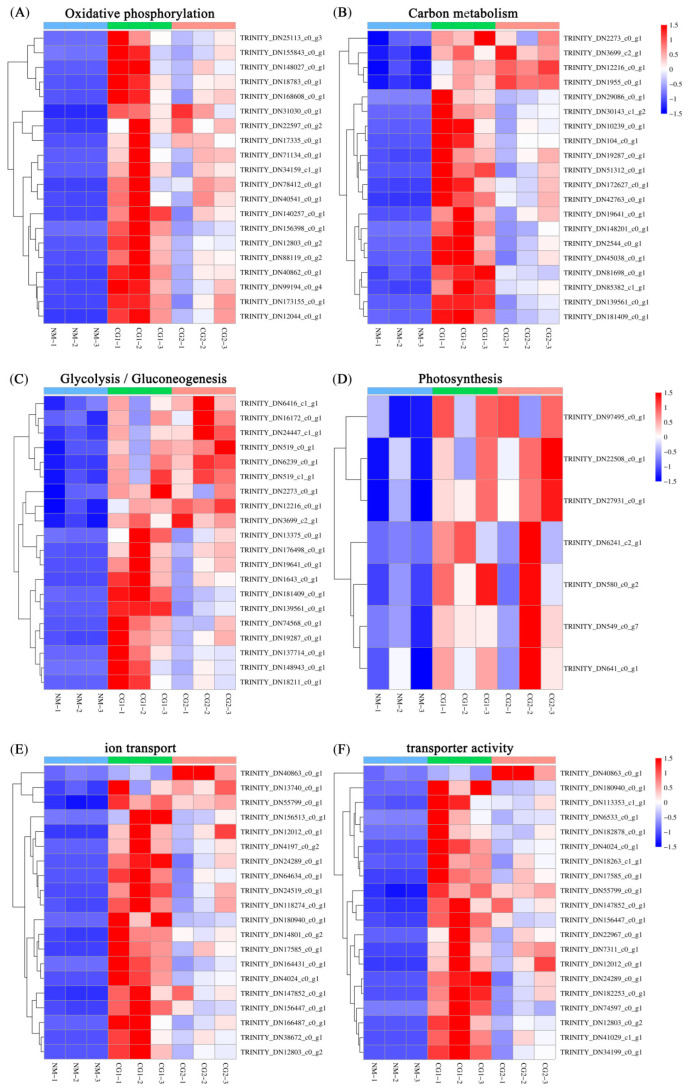
The heatmap of part up-regulated DEGs in the different pathways in NM, CG1, and CG2 treatments. (**A**) oxidative phosphorylation pathway, (**B**) carbon metabolism pathway, (**C**) glycolysis/gluconeogenesis pathway, (**D**) photosynthesis pathway, (**E**) ion transport pathway, (**F**) transporter activity pathway. The Y- and X-axes represent the differentially expressed genes and different samples. The different colors of the heatmap, ranging from blue to white to red, represent scaled expression levels of genes with [log10(FPKM + 0.000001)] across different samples.

**Figure 12 plants-12-03179-f012:**
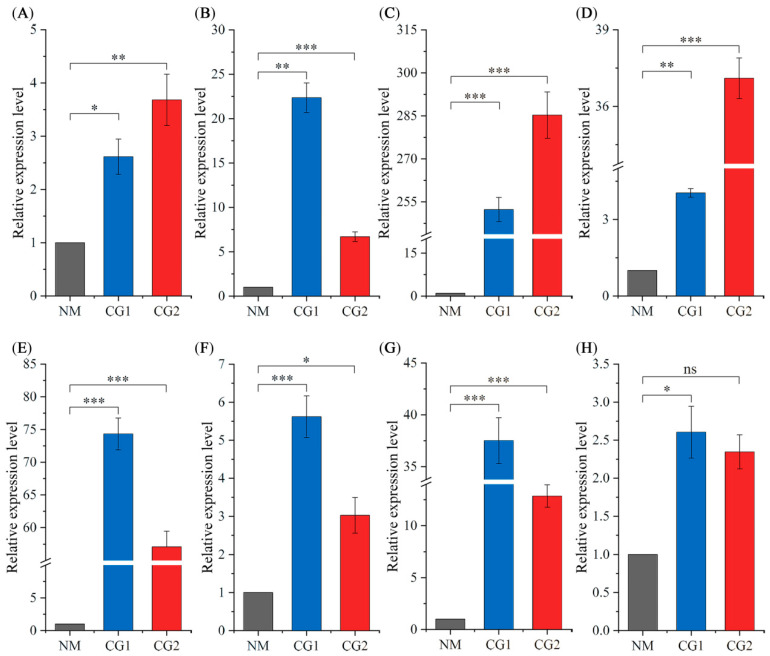
RT-qPCR analysis of the expression level of eight candidate genes in different samples. (**A**) TRINITY_DN81698_c0_g1, (**B**) TRINITY_DN156398_c0_g1, (**C**) TRINITY_DN22597_c0_g2, (**D**) TRINITY_DN40863_c0_g1, (**E**) TRINITY_DN55799_c0_g1, (**F**) TRINITY_DN4024_c0_g1, (**G**) TRINITY_DN74597_c0_g1, and (**H**) TRINITY_DN17585_c0_g1. The relative expression changes of the target gene in NM, CG1, and CG2 treatments were calculated by the 2^−∆∆Ct^ method, and are shown as means ± SE (n = 3). The statistically significant difference between NM and CG1/CG2 treatments is tested by Student’s t-test (* *p* < 0.05; ** *p* < 0.01; *** *p* < 0.001, ns indicates no significant difference).

**Table 1 plants-12-03179-t001:** Root morphological index.

Treatment	Length	SurfArea	AvgDiam	RootVolume	Tips	Forks
NM	174.1 ± 8.88 a	24.91 ± 1.51 b	0.46 ± 0.02 b	0.28 ± 0.03 b	323 ± 5.13 a	831.33 ± 21.46 b
CG1	197.72 ± 20.48 a	40.22 ± 4.91 a	0.65 ± 0.02 a	0.66 ± 0.11 a	359.33 ± 69.29 a	1353.67 ± 83.56 a
CG2	208.68 ± 28.84 a	38.63 ± 3.06 a	0.69 ± 0.09 a	0.62 ± 0.08 a	337.33 ± 26.49 a	989.33 ± 96.53 b

Total root length, root surface area, average root diameter, root volume, and root tips and root forks numbers of non-inoculated (NM) and inoculated (CG1, CG2) *Massion’s pine* plants grown in toxic-metal-contaminated mine soil. The average values ± SE of three replicates. Different letters indicate statistically significant differences between different *C. geophilum* inoculations using one-way ANOVA followed by Dunnett’s test (*p* < 0.05).

**Table 2 plants-12-03179-t002:** Soil physical and chemical properties.

Treatment	pH	TK (g·kg^−1^)	TP (g·kg^−1^)	SOM (g·kg^−1^)	Cr (g·kg^−1^)	Pb (g·kg^−1^)	Mn (g·kg^−1^)	Cd (mg·kg^−1^)
OS	6.21 ± 0.17	5.22 ± 0.29	0.51 ± 0.02	26.73 ± 0.76	0.37 ± 0.02	0.78 ± 0.01	22.76 ± 1.38	12.90 ± 0.26
NM	5.27 ± 0.03 a	4.22 ± 0.05 a	0.61 ± 0.01 a	27.87 ± 1.37 a	0.35 ± 0.01 a	0.74 ± 0.02 a	20.4 ± 1.37 a	2.87 ± 0.42 b
CG1	5.48 ± 0.09 a	4.17 ± 0.09 a	0.55 ± 0.07 a	26.12 ± 2.24 a	0.34 ± 0.01 a	0.63 ± 0.02 b	18.73 ± 0.65 a	3.65 ± 0.77 b
CG2	5.49 ± 0.1 a	4.44 ± 0.42 a	0.58 ± 0.08 a	29.09 ± 1.06 a	0.32 ± 0.02 a	0.70 ± 0.01 b	20.8 ± 2.03 a	5.76 ± 0.38 a

Contents of pH, TK, TP, SOM, Cr, Pb, Mn, and Cd in mine soil (OS) in rhizosphere soil of non-inoculated (NM) and inoculated (CG1, CG2) *Massion’s pine* plants. The average values ± SE of three replicates. Different letters indicate a statistically significant difference between different ECM fungal inoculations by one-way ANOVA analysis followed by Dunnett’s test (*p* < 0.05). OS represents original soil.

## Data Availability

The data presented in this study are available within the article and Appendix A. Data for *Massion’s pine* seedlings roots raw sequence reads are available in a publicly available repository (NCBI), reference number PRJNA987420.

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
