# Peer review of "Increased Tolerance of Massion’s pine to Multiple-Toxic-Metal Stress Mediated by Ectomycorrhizal Fungi"

_plants, 2023, doi:10.3390/plants12183179_

Round 1
Reviewer 1 Report
The manuscript deals with the assessment of Pinus massoniana response to heavy metal stress when exposed to mycorrhizal fungi Cenococcum geophilum. The manuscript would probably be more interesting if tables and figures were included. The details are listed below:
L21-23: add some % changes of examined parameters between control and inoculated treatments
L74-76 and L78-80: these statements have opposite meaning
L121: indicate how much soil and HClO4 and HF were used
L127: remove .re
L131: how much water and how often the seedlings were irrigated?
L148: describe the procedure for the determination of osmotic regulating substances. What compounds were analyzed?
L153-155: briefly describe the procedure for the determination of antioxidant enzymes activity
L157, 162: indicate a volume of mixed solution and sulfuric acid
L199: ‘NM’ – full name
L238: CO2
L244: add a procedure in Materials and methods for proline determination
L261: ‘NM vs. CG1’, ‘NM vs. CG2’ – what does it mean?
L260-263: indicate clearly how many genes were upregulated and downregulated in CG1 and CG2 compared to NM
L307: indicate the species name of EMF
L316: Salix in italics
L316-318, 334: indicate the species name for the first time used
L331: Suillus luteus
L381-383: for comparison indicate the plant response to pathogenic fungi. For example biotic stress caused by Botrytis cinerea in lettuce induced the activity of CAT, SOD and glutathione, while Fusarium culmorum in wheat induced the activity of CAT and POD. For this purpose the Authors may refer to the following references: https://doi.org/10.1007/s00425-022-03838-x, https://doi.org/10.3390/agronomy13051378
L385: ‘Consistent with previous work’ – indicate this study
L395: ‘oxidative injury’ – rephrase
L397-399: add a mode of action of MDA: reduced membrane lipid peroxidation by CG1 and CG2
L414-415: ‘enzyme activities of the antioxidant enzymes’ - rephrase
Moderate editing of English language required
Author Response
“Please see the attachment”

Reviewer 2 Report
The authors of the manuscript titled “Increased tolerance of Masson’s pine to multiple-heavy metal stress mediated by ectomycorrhizal fungi” current study aimed to investigate the contribution of EMF on the growth, heavy mental accumulation, absorption of mineral nutrients, antioxidant capacity and related gene expression of Mason’s pine exposed to a multiple-metal contaminated soil.
General comments
Overall, the study is well-designed and presented in a good way.
Keywords
Authors are advised to carefully review the keywords and make necessary corrections.
Title
The title of the study seems acceptable.
Abstract
When discussing the effects of heavy metals and fungal inoculation on plant growth and crop production, authors should acknowledge any gaps in current knowledge or limitations. It's important to provide specific percentages of increase or decrease and explain how they relate to the physiological or biochemical mechanisms that regulate drought tolerance, rather than simply stating general results. This highlights the global significance of addressing this issue.
Introduction
The authors are requested to rewrite the objectives of the study and make it easy for the readers.
Results
The authors made several mistakes when indicating significant differences using symbols. As a result, they are being asked to reorganize the graphs and conduct again test to measure significance.
Discussion
To enhance the study, we kindly request the authors to provide numerical data on the parameters examined. Moreover, it would be helpful if the authors could highlight any obstacles and limitations faced during the research and suggest potential solutions for such challenges.
Conclusion
This section of the manuscript appears to be satisfactory.

Author Response
“Please see the attachment”

Reviewer 3 Report
Please see my comments about the corresponding MS (Plants-2538787)
The manuscript entitled “Increased tolerance of Massion’s pine to multiple-heavy metal stress mediated by ectomycorrhizal fungi” aims to address the effect of Ectomycorrhiza fungi (EMF), on the growth, metal accumulation, absorption of mineral nutrients, antioxidant capacity and related gene expression of Massion’s pine exposed to a multiple-metal contaminated soil.
I think this research has certain innovations. Further, I think that the topic content of this manuscript is also suitable for the international audiences of Plants. I encourage the authors to review and modify carefully the manuscript and after checking the below points I am pretty sure it should be accepted for publication:
-First, I would like to underline that the use of the term "heavy metals", which was never internationally defined, should be discouraged. The term "heavy metals" should be replaced by "trace elements" (TE), "metals" "trace metals", "toxic elements" (for metal whose toxicity is proven) or "potentially toxic elements" throughout the text.
-Keywords should not include words that are already mentioned in the title.
-It is not clear how (why) studied elements (Mn, Cr, Cd and Pb) were selected (Even the selection is not bad). Heavy metals affecting plants need to be more specified.
- Inhibition of plant growth by reducing nutrient uptake by plants has been addressed by several author such as Jeddi et al (Jeddi, K., Siddique, K. H., Chaieb, M., & Hessini, K. (2021). Physiological and biochemical responses of Lawsonia inermis L. to heavy metal pollution in arid environments. South African Journal of Botany, 143, 7-16.). These references and other could be used in the Introduction section.
-The materials and methods section needs further precisions (particularly, the significance of biological factors must be mentioned).
- Results section: Overly descriptive.
-This is a short term metal stress application, a short term seed inoculation and short term physiological and molecular Massion’s pine response. The experiments were carried out on very young seedlings, with no very developed leaves. The authors in the discussion section must take into consideration these experimental conditions.
- Discussion section: Please improve the readability as far as possible. Further, the authors must fully elaborate the most significant results. The depiction in the discussion section must be closely connected to the results of this manuscript and the results of previous investigators.
- The manuscript lack a paragraph to show what is the important of this study to engineers in future study?
- Finally: The manuscript is need to improving in English language, please improve it.
Moderate editing of English language required
Author Response
“Please see the attachment”

Reviewer 4 Report
Review report ID Plants-2538787:
Summary
Title: Increased tolerance of Massion’s pine to multiple-heavy metal stress mediated by ectomycorrhizal fungi
The manuscript investigates the role of Ectomycorrhizal fungi (EMF) in mediating heavy metal tolerance in host plants. Massion’s pine seedlings, the plant model in this study, were inoculated with two strains of Cenococ cum geophilum (C. geophilum) (isolated from a polluted mine area). The authors claim that EMF accelerated the seedling growth, element absorption, photosynthesis, and decreased Cr, Cd, Pb and Mn accumulation in shoots. C. geophilum inoculation boosted proline content and antioxidant enzymes CAT, POD and SOD activities in shoots and roots of Massion’s pine seedlings. Comparative transcriptomic analysis demonstrated that differently expressed genes (DEGs) were mainly encoding functions involved in “transmembrane transport”, “ion transport”, “oxidation/reduction processes”, “oxidative phosphorylation”, “carbon metabolism”, “glycolysis/gluconeogenesis”, “photosynthesis” and “bio synthesis of amino acids”. Finally, the authors claim that these results indicate that C. geophilum mitigates heavy mental stress by promoting nutrient uptake, photosynthesis and plant growth, thereby modulating the plant antioxidant system to reduce oxidative stress, and reducing the transport and enrichment of heavy metals from the root to the shoot of Massion’s pine seedlings.
General concept comments
The Article writing could benefit from some improvements. This is particularly important regarding figure´s captions. In general, figures captions are not clear enough for the reader making the review process quite challenging. In several instances captions don´t state properly what is displayed in the figure. As example consider the following:
Lines 219 to 223
Figure 1. The biomass of Pinus massoniana inoculated with Cenococcum geophilum (CG1,CG2) and non-inoculated plants (NM) under mine soil. A Seedling height, B shoot fresh weight (SFW), root fresh weight (RFW), shoot dry weight (SDW), root dry weight (RDW). Bars represent average values±SE of 3 replicates. Different letters indicate significant differences among different ECM fungal treatments using one-way ANOVA followed by Dunnett’s test (P<0.05).
Figure1. Effect of Cenococcum geophilum C1 and C2 inoculation on seedling height (A) and plant biomass (B) of Pinus massoniana grown in heavy metal contaminated mine soil for “X” days. Bars represent average values ± SE of 3 replicates. Different letters indicate statistically significant difference between different ECM fungal inoculations using one-way ANOVA followed by Dunnett’s test (P<0.05). Non-inoculated plants (NM). Shoot fresh weight (SFW). Root fresh weight (RFW). Shoot dry weight (SDW). Root dry weight (RDW).
An English revision would be also needed to improve the clarity of the text.
This study contributes with a very interesting approach regarding antioxidant activities and their related gene expression in Massion’s pine plants exposed to a multiple-metal contaminated soil and has importance from the ecological point of view.
Minor comments
Line 20. Across the paper we find Ectomycorrhizal fungi (EMF) and (ECM), for clarity could you use only one acronym. Please review and edit accordingly.
Line 28. Please correct “mental”.
Table 1. Lines 229-230. Could you indicate the meaning of CK? Additionally, the length of the plant root under different treatments has a reduction of, at least 11%, but there is no indication of statistically significant difference, could you elaborate on this?
Figure 2 caption. Please edit for clarity.
Line 246, 247,252,253. Please replace comas “、” for “,”.
Line 246-247. Redundant sentence. “Compared to NM seedlings, Cr、Cd、Pb and Mn uptake was significantly different between mycorrhizal and non-mycorrhizal seedlings.”
Please consider an edition like the following: Compared to NM seedlings, Cr, Cd, Pb and Mn uptake was significantly different in inoculated plants.
Lines 257-261. Please edit figure 3 captions for clarity and pay attention to mistakes like Figure3.
Line 263. Could you elaborate on the TF concept and how is calculated? it could be mentioned on M&M.
Lines 266-268. For clarity please consider:
Compared to NM, the transport of Cr, Cd, Pb and Mn from root to shoot was reduced by 32%, 26%, 75%, 64%, and by 58%, 17%, 68%, and 18%, respectively, in CG1 and CG2 inoculated seedlings (Figure 3C).
Lines 287-296. For clarity, please edit figure 4 and 5 captions.
Please consider a similar edition to this:
Figure 5. Malondialdehyde (MDA) (A) and Proline (Pro) (B) content in shoot and roots of non-inoculated (NM) and inoculated (CG1, CG2) Pinus massoniana plants grown in heavy metal contaminated mine soil. Bars represent average values ± SE of 3 replicates. Different letters indicate statistically significant difference between different ECM fungal inoculation by one-way ANOVA analysis followed by Dunnett’s test (P < 0.05).
Lines 309-313. For clarity, carefully review figure caption, edit text when need it, and correct errors. For instance, Figure 6 instead of Figure6.
Line 315. 55.83 GB
Lines 328-330. For clarity, carefully review figure caption, edit text when need it, and correct errors. For instance, Figure 7 instead of Figure7.
Lines 332-334. For clarity, carefully review figure caption, edit text when need it, and correct errors. For instance, Figure 8 instead of Figure8.
Line 335. There is an inconsistency on gene ontology acronym (Go and GO) please correct it. Please, when first mentioned on the text, indicate the meaning of GO analysis.
Lines 335-344. The paragraph is not clear for the reader. Could you please edit it?
Line 340. Please correct “orangelle”.
Lines 346-349. For clarity, carefully review figure caption, edit text when need it, and correct errors. For instance, biological progress. Please correct errors on the figure as well. Please indicate the meaning of “establishment of localization” and “localization” on the figure.
Lines 358-361. For clarity, carefully review figure caption, edit text when need it, and correct errors. For instance, Figure 10 instead of Figure10. When first mentioned on the manuscript´s main text, clarify the meaning of KEGG.
Line 377. Please replace genes by gene.
Lines 380-381. RT-qPCR stands for reverse transcription-quantitative polymerase chain reaction.
Lines 382-383. I cannot understand the meaning of the sentence “The qPCR transcriptome data exhibited a consistent expression that eight genes were significantly up-regulated (Figures 12A-H).” Could you please edit it or rephase it for clarity?
Lines 387-393. Could you please edit Figure 12 captions for clarity?
Lines 417-422. The paragraph is very difficult to follow. Could you edit it for clarity?
Line 448. Please replace Go by GO.
Lines 451-456. The paragraph is very difficult to follow. Could you edit it for clarity?
Lines 462, 468 and 469. “Phosphorilation”, “oxidative phosphorylation” and “carbon metabolism” have different font type and/or different size. Could you correct it and make sure to check the entire manuscript for this?
Line 489. Please correct “seelding”
Line 491. Please replace “injury” by damage.
Line 495. Please correct “seedings”.
Mayor comments
Line 124. Table S1, which is not available, is critical for the review of this work. Please provide the table.
Lines 201-207. The authors claim that: “The results showed that CG1 and CG2 could promote the plant biomass under heavy metal stress. The heavy metal stress in the mine soil significantly inhibited the root development, but the root surface area, average root diameter, root volume, number of root tips, and number of bifurcations of Massion’s pine seedlings inoculated with C. geophilum were significantly higher than NM seedlings (p<0.05), indicating that C. geophilum can effectively promote the root development and improve the tolerance of Massion’s pine to heavy metals (Table 1).”
However, it is impossible to assess this claim because we don´t know the concentration of heavy metals in soil (table S1 not provided) and additionally, there is no experimental control to differentiate between heavy metal negative effect on the plant and the positive effect of the EMF. The analysis requires a non-contaminated soil to grow non inoculated plants in order to assess properly the effect of HM and inoculation on the plant.
Could you please provide the required table and an explanation for the missing experimental control? Have the authors done previous work showing the negative effect of HM contaminated soil on this plant?
Lines 298-300. The authors claim that “C. geophilum substantially increased proline content in shoots and roots of Massion’s pine seedlings under heavy metal stress, and the content of proline in the shoot was higher than that in the root.” However, evidences of the plants growing under heavy metal stress is not provided. Additionally, Figure 1A shows that seedlings height of non-inoculated plants is similar to those inoculated, and Figure 1B shows that shoot dry weight of non-inoculated plants is similar to those non-inoculated, indicating low abiotic stress. This also indicates that the difference on fresh weight could be due to water content, suggesting that the differences we see on biomass are due to the beneficial effect of the inoculation instead of the alleviation of heavy metal stress.
Could you provide an explanation for this?
Lines 297-302. There is no mention of figure 5A on the manuscript text.
Line 316 and 318. Table S3 and S4 are not available.
Lines 350-356. The authors claim that “We mapped the DEGs to the reference canonical pathways in KEGG to further identify the regulatory pathways of these DEGs involved in alleviating heavy metal toxicity to plants mediated by C. geophilum (Figures 10A, B). CG1 and CG2 shared the pathways of “oxidative phosphorylation”, “carbon metabolism”, “glycolysis/gluconeogenesis”, “photosynthesis” and “biosynthesis of amino acids”, indicating that nutrient and energy utilization is important response mechanisms of Massion’s pine seedlings to heavy mental 355 stress mediated by C. geophilum occurred.” However, there is no available evidence of heavy metal stress, hence the identified pathways could be related solely to the beneficial effect of inoculation.
Lines 363-365. The authors claim that “We created a heatmap of part up-regulated DEGs enriched in the same pathway to select the key genes of Massion’s pine conferring resistance to heavy mental stress mediated by C. geophilum“. However, the heatmap could indicate correlation between pathways and tolerance to HM only if evidences of HM stress are provided. The correlation is not evidence of causing tolerance or resistance to HM, nor the upregulation of gene expression. The downregulation of pathways involved in the expression of transport of cations could be correlated as well with tolerance to HM. Could you comment on this?
The Article writing could benefit from some improvements. This is particularly important regarding figure´s captions. In general, figures captions are not clear enough for the reader making the review process quite challenging.
An English revision would be also beneficial to improve the clarity of the text, specifically in results and discussion sections of the article.
Author Response
“Please see the attachment”

Round 2
Reviewer 1 Report
The Authors have improved the paper significantly. However, as it was indicated previously some contrasting information related to the impact of pathogenic fungi on plant metabolism is needed in L551-554. For example biotic stress caused by Botrytis cinerea in lettuce induced the activity of CAT, SOD and glutathione, while Fusarium culmorum in wheat induced the activity of CAT and POD. For this purpose the Authors may refer to the following references: https://doi.org/10.1007/s00425-022-03838-x, https://doi.org/10.3390/agronomy13051378
